# Stabilization of PIM Kinases in Hypoxia Is Mediated by the Deubiquitinase USP28

**DOI:** 10.3390/cells11061006

**Published:** 2022-03-16

**Authors:** Rachel K. Toth, Regina Solomon, Noel A. Warfel

**Affiliations:** 1University of Arizona Cancer Center, Tucson, AZ 85724, USA; racheltoth@arizona.edu; 2Department of Biochemistry, Cell & Molecular Biology, The University of Texas Medical Branch at Galveston, Galveston, TX 77555, USA; regina_solomon_5g@yahoo.com; 3Department of Cellular and Molecular Medicine, University of Arizona, Tucson, AZ 85724, USA

**Keywords:** PIM kinases, USP28, hypoxia

## Abstract

Proviral integration sites for Moloney murine leukemia virus (PIM) kinases are upregulated at the protein level in response to hypoxia and have multiple protumorigenic functions, promoting cell growth, survival, and angiogenesis. However, the mechanism responsible for the induction of PIM in hypoxia remains unknown. Here, we examined factors affecting PIM kinase stability in normoxia and hypoxia. We found that PIM kinases were upregulated in hypoxia at the protein level but not at the mRNA level, confirming that PIMs were upregulated in hypoxia in a hypoxia inducible factor 1-independent manner. PIM kinases were less ubiquitinated in hypoxia than in normoxia, indicating that hypoxia reduced their proteasomal degradation. We identified the deubiquitinase ubiquitin-specific protease 28 (USP28) as a key regulator of PIM1 and PIM2 stability. The overexpression of USP28 increased PIM protein stability and total levels in both normoxia and hypoxia, and USP28-knockdown significantly increased the ubiquitination of PIM1 and PIM2. Interestingly, coimmunoprecipitation assays showed an increased interaction between PIM1/2 and USP28 in response to hypoxia, which correlated with reduced ubiquitination and increased protein stability. In a xenograft model, USP28-knockdown tumors grew more slowly than control tumors and showed significantly lower levels of PIM1 in vivo. In conclusion, USP28 blocked the ubiquitination and increased the stability of PIM1/2, particularly in hypoxia. These data provide the first insight into proteins responsible for controlling PIM protein degradation and identify USP28 as an important upstream regulator of this hypoxia-induced, protumorigenic signaling pathway.

## 1. Introduction

Hypoxia is common in cancers. As the tumor proliferates, it rapidly outgrows its blood supply, leading to areas of low oxygen tension. Both healthy and tumor cells compensate for low oxygen tension by enacting a transcriptional program driven by the hypoxia-inducible factor 1 (HIF-1) transcription factor [1]. However, tumor cells have additional adaptive responses to hypoxia that allow them to survive in this harsh microenvironment [2]. Identifying such factors, particularly if they are actionable targets, may provide potential therapeutic options to oppose the well-established oncogenic effects of hypoxia in patients with solid tumors.

The Proviral Integration site for Moloney murine leukemia virus (PIM) proteins are serine/threonine kinases that are involved in cytokine signaling [3]. They are upregulated in multiple cancer types, most commonly in hematopoietic cancers and prostate cancer [4,5,6,7]. PIM kinases are best known for their role in helping cells to evade apoptosis through their direct phosphorylation of BAD [8,9,10]. However, they also promote tumor cell survival through other mechanisms, such as decreasing lethal levels of reactive oxygen species and regulating mitochondrial dynamics [11,12]. PIM kinases have also been implicated in other hallmarks of cancer, including regulation of cellular energetics [13], promoting tumor angiogenesis [14], helping tumor cells evade the antitumor immune response [15], and promoting cell motility [16]. Despite this multifaceted role in cancer progression, very little is known about how PIM protein stability is controlled. This is especially poignant, because, unlike other kinases, PIM kinases do not possess regulatory domains [17]. Most kinase cascades are activated or repressed through post-translational modification, which allows for the rapid control of signal transduction. However, PIM kinases contain no regulatory domains that would allow for rapid shutoff [17], and very few phosphorylation sites have been identified [18,19]. Therefore, it is thought that once PIM kinases are transcribed and translated, they are constitutively active. This is evident in the crystal structure of PIM1, which indicates that PIM1 has a tertiary structure similar to those of other constitutively activated kinases and the ability to bind an ATP analog in the absence of phosphorylation [17].

PIM kinases are activated transcriptionally, most commonly through the Janus kinase (JAK)/signal transducer and activator of transcription (STAT) signaling [20] and nuclear factor-κB signaling [21]. PIM transcripts tend to be short-lived because of their unstable 3′-untranslated regions [22]. PIM kinase transcripts, particularly *PIM1* and *PIM3*, have also been shown to be regulated by microRNAs in cancer [23,24,25]. However, there has been little research regarding regulation of PIM kinases at the protein level. PIM1 was reported to bind to heat shock protein (HSP) 70 and HSP90, which enhances and hinders its stability, respectively [26]. Protein phosphatase 2A has also been shown to regulate PIM stability, although the mechanisms have not been described [27]. While observations in the literature indicate that PIM kinases are likely degraded by the 26S proteasome [28], the factors that regulate PIM ubiquitination and degradation remain unknown. Therefore, understanding how PIM kinases are regulated at the protein level is vital for understanding their role in cancer and effectively targeting them therapeutically.

PIM kinases are strongly induced by hypoxia [29,30]. The majority of hypoxia-induced proteins are targets of the HIF-1 transcription factor. However, the levels of *PIM1/2* do not increase in response to hypoxia, suggesting that PIM is upregulated at the protein level in hypoxia, independent of HIF-1-mediated transcription [11]. The majority of protein degradation in the cell occurs at the proteasome. Proteins that have been polyubiquitinated are shuttled to the 26S proteasome for degradation. Polyubiquitination is catalyzed by a series of ubiquitin ligases, the last of which—the E3 ligase—is highly specific for the target protein. This process is reversible, and deubiquitinases (DUBs) are the proteins responsible for removing ubiquitin marks from proteins, thus saving them from degradation by the proteasome [31]. Therefore, DUBs are vital players in the regulation of protein stability. There are approximately 100 DUBs in the human genome, and multiple DUBs have been implicated in cancer [32]. DUBs may be particularly useful in hypoxia, as signaling pathways that expend too much energy need to be rapidly shut down. Proteins that might otherwise be targeted for degradation may become more stable, as the process of translation requires energy that is unavailable in low oxygen conditions. Several DUBs are known to be differentially active in hypoxia [32]. One such DUB is ubiquitin-specific protease (USP) 28. USP28 is of particular interest because it has been shown to regulate the stability of HIF-1α in hypoxia [33], although it likely also has other hypoxia-specific targets. USP28 has been described to have both tumor-suppressive and oncogenic roles, depending on the cellular context [34]. In addition to regulating HIF-1α, USP28 has been shown to regulate cell cycle regulators, such as checkpoint kinases [35], and tumor protein 53-binding protein 1 [36,37], thereby promoting cell cycle arrest and apoptosis. In addition, USP28 has also been shown to deubiquitinate the proto-oncogene MYC [38], and PIM kinases have been previously shown to cooperate with MYC to promote tumorigenesis [39,40].

In this study, we are the first to identify USP28 as a critical factor regulating the stability of PIM kinases. In hypoxia, USP28 interacts with and deubiquitinates PIM1 and PIM2, resulting in increased protein stability and signaling to PIM substrates. The knockdown of USP28 blunts the upregulation of PIM1 in xenograft tumor models and results in reduced tumor growth.

## 2. Materials and Methods

### 2.1. Tissue Culture

HCT116 (human colon cancer) and 293T (transformed human embryonic kidney) cells were purchased from the American Type Culture Collection (Manassas, VA, USA). PC3-LN4 (prostate cancer) cells were a gift from Dr. Andrew Kraft. This cell line was created from the serial orthotopic transplantation of parental PC3 cells and subsequent harvest from lymph node metastases [41]. Parental and genetically modified HCT116 and PC3-LN4 cells were grown in RPMI with 10% fetal bovine serum (FBS), and 293T cells were grown in DMEM with 10% FBS. All cells were cultured at 37 °C in 5% CO_2_, routinely screened for mycoplasma, authenticated by short tandem repeat DNA profiling by the University of Arizona Genetics Core Facility, and used for fewer than 50 passages. For experiments involving hypoxia (1% O_2_), cells were cultured in a hypoxic environment (1% O_2_, 5% CO_2_, and 94% N_2_) using an InVivo2 400 hypoxia workstation (Baker Ruskinn, Sanford, ME, USA).

PC3-LN4-knockdown cells were created by transducing cells with a control virus or viruses encoding short hairpin RNAs (shRNAs) against USP28. Cells were selected under puromycin treatment.

### 2.2. Plasmids and Reagents

Flag-USP13, Flag-USP28, HA-CYLD lysine 63 deubiquitinase (CYLD), HA-USP46, pRetroSuper, pRetro-shUSP28-1, and pRetro-shUSP28-3 plasmids were purchased from Addgene (Watertown, MA, USA). HA-PIM1 and HA-PIM2 were gifts from Dr. Andrew Kraft. USP28-Myc, GFP-PIM3, and Flag-F-box and WD repeat domain-containing 7 (FBW7) were created using Gateway cloning. Cells were cultured to approximately 70% confluency, at which point plasmids were transfected using Lipofectamine 3000 (Thermo Fisher, Waltham, MA, USA).

MG-132, PR-619, and AZD5383 were purchased from Selleck Chemical (Houston, TX, USA). HBX 41108 was purchased from Cayman Chemical (Ann Arbor, MI, USA). Cycloheximide (CHX) was purchased from VWR (Randor, PA, USA).

### 2.3. Quantitative Polymerase Chain Reaction (qPCR)

RNA was extracted using the Quick-RNA Miniprep Kit (Zymo, Irvine, CA, USA) and reverse-transcribed using the qScript cDNA Synthesis Kit (Quantabio, Beverly, MA, USA). qPCR was performed using a CFX96 Lightcycler (Bio-Rad, Hercules, CA, USA) and qPCRBIO qPCR Master Mix (PCR Biosystems, Wayne, PA, USA). Primers were purchased from Qiagen (Germantown, MD, USA) or Integrated DNA Technologies (Coralville, IA, USA). Primer sequences are listed in Table 1. The expression of target genes relative to beta-actin was quantified using the 2^−^^ΔΔCT^ method.

### 2.4. Western Blotting

Proteins were extracted from cells or tumor tissues using RIPA buffer (150 mM NaCl, 1% NP-40, 0.5% sodium deoxycholate, 0.1% sodium dodecyl sulfate, and 50 mM Tris, pH 7.4) with protease inhibitors, and an equal amount of each lysate was loaded onto a 10% sodium dodecyl sulfate-polyacrylamide gel. Lysates were electrophoretically transferred to polyvinylidene fluoride or nitrocellulose membranes, which were then blocked with 5% milk or 1% casein, respectively. Membranes were washed with Tris-buffered saline with Tween (TBST) and incubated with primary antibodies at 4 °C overnight. Membranes were then washed with TBST and incubated with horseradish peroxidase-conjugated or fluorescently labeled secondary antibodies for 1 h at room temperature and imaged using ECL or a LiCor imager, respectively.

The following primary antibodies were purchased from Cell Signaling (Beverly, MA, USA): HA, Flag, PIM1, PIM2, PIM3, HIF1α, Myc, GFP, phospho-glycogen synthase kinase 3β (pGSK-3β), and ubiquitin. The USP28 antibody was purchased from Abcam (Beverly, MA, USA). Actin (BD Biosciences, Franklin Lakes, NJ, USA) was used as a loading control.

### 2.5. Immunoprecipitation and Protein Degradation Assays

To assess the ubiquitination of PIM isoforms under different conditions, 293T cells or PC3-LN4 USP28-knockdown cells were transfected with HA-tagged PIM1, PIM2 (HA-PIM1 or HA-PIM2), or GFP-PIM3 overnight. Then, as appropriate, cells were treated with PR-619 or placed in hypoxia. Cells were treated with the proteasome inhibitor MG-132 for 2 or 4 h before harvest to block the degradation of ubiquitinated PIM isoforms. To determine whether USP28 bound to PIM kinases, 293T cells were co-transfected with Myc-tagged USP28 (USP28-Myc) and HA-tagged PIM1 or PIM2 (HA-PIM1 or HA-PIM2) overnight. Then, cells were treated as stated and placed in hypoxia for the stated times. Cells were harvested in an IP lysis buffer (20 mM Tris HCl, pH 8; 137 mM NaCl; 10% glycerol; 1% Nonidet P-40; and 2 mM EDTA) with protease inhibitors and centrifuged at 15,000 RPM for 10 min. Lysates were incubated overnight at 4 °C with HA magnetic beads (Pierce Biotechnology, Waltham, MA, USA) or GFP magnetic beads (Chromotek, Islandia, NY, USA) and subjected to western blotting as described above.

To assess changes in protein degradation, 293T cells were co-transfected with HA-PIM1 and either Flag-USP28 or a control vector. The following day, cells were treated with 10 µM CHX in hypoxia or normoxia to block new translation and harvested at the stated time points. Western blotting was performed as described above.

### 2.6. In Vivo Experiments

All animal studies were approved by the Institutional Animal Care and Use Committee of the University of Arizona. Male NOD/SCID mice at 6–8 weeks of age were used. Five million control or shUSP28-1 PC3-LN4 cells in PBS were injected subcutaneously into the rear flanks of eight mice each. Tumor volume was measured over time by caliper and calculated using the equation: V = (tumor width)^2^ × tumor volume/2. Mice were administered sunitinib (100 mg/kg; Adooq Bioscience, Irvine, CA, USA) or vehicle daily, once the tumors reached ~100 mm^3^ (*n* = 4 mice and 8 tumors/group). Mice were sacrificed when the tumor volume reached ~2000 mm^3^. After sacrifice, tumors were harvested for downstream experiments. Immunohistochemical staining was performed to assess PIM1 levels.

### 2.7. Statistical Analysis

Western blot densitometry was performed using Image J v1.51u (National Institutes of Health, Bethesda, MD, USA). Statistical analysis was performed using Microsoft Excel, version 2108 (Microsoft, Redmond, WA, USA). A *p* value < 0.05 was considered statistically significant.

## 3. Results

### 3.1. PIM Kinases Are Upregulated in Hypoxia at the Protein Level

We have previously observed that PIM kinases are increased in hypoxia in prostate, breast, and colon cancer cells [14]. However, the mechanism underlying this increase in PIM protein levels is unknown. Many proteins that are upregulated in hypoxia are target genes of the hypoxia-inducible transcription factor HIF-1. To assess whether the PIM isoforms are transcriptionally upregulated in hypoxia, we examined the protein and RNA levels in cells cultured in hypoxia (1.0% O_2_) for 4 or 8 h. In both the HCT116 and PC3-LN4 cell lines, we observed robust increases in PIM1, PIM2, and PIM3 protein levels after 4 and 8 h in hypoxia compared to the levels in normoxia (Figure 1A). Notably, although classic HIF-1 target genes, such as hexokinase 2 (*HK2*), were increased in hypoxia, there were no significant differences in the mRNA levels of *PIM1*, *PIM2*, or *PIM3* (Figure 1B) in hypoxia and normoxia, indicating that the increase in PIM kinase levels in hypoxia occurs at the post-translational level and is independent of HIF-1 transcriptional activation.

The total levels of a majority of cellular proteins were regulated through degradation by the 26S proteasome, which occurred after polyubiquitination. To determine whether hypoxia altered the rate of PIM ubiquitination, we transfected HA-PIM1, HA-PIM2, or GFP-PIM3 into 293T cells that were cultured in normoxia or 1.0% O_2_ prior to treatment with MG-132 (10 μM), a proteasome inhibitor, to block proteasomal degradation and preserve the ubiquitinated form of PIM. Lysates were collected over time, the tagged proteins were immunoprecipitated, and ubiquitination was assessed by immunoblotting. The rates and total amounts of ubiquitination of all PIM isoforms were significantly reduced in hypoxia compared to those in normoxia after 2 and 4 h of MG-132 treatment (Figure 1C), suggesting that the ubiquitination of PIM kinases is impaired in hypoxia compared to in normoxia, which favors protein stability. Because PIM1 and PIM3 frequently act on similar substrates and have high homology [42,43], we expect that PIM1 and PIM3 are regulated similarly. Therefore, we focused on the regulation of PIM1 and PIM2 for further experiments.

### 3.2. PIM Kinases Are Regulated by DUBs

Decreased ubiquitination can result from either the decreased activation of E3 ubiquitin ligases or increased activation of DUBs. To determine whether PIM levels are sensitive to deubiquitination, we treated PC3-LN4 cells with PR-619, a pan-DUB inhibitor, or HBX 41108, a USP7 inhibitor that has shown broader spectrum activity at low concentrations [44]. The treatment with PR-619 significantly decreased PIM1 and PIM2 protein levels, whereas HBX 41108 had no effect on PIM levels (Figure 2A). These data indicated that PIM1/2 stability was acutely controlled by deubiquitination. As expected, the treatment with MG-132 increased PIM1 and PIM2 levels, confirming that PIM kinases were degraded by the 26S proteasome. Notably, MG-132 treatment blocked the reduction of PIM levels observed with PR-619. Together, these data indicated that deubiquitination plays a key role in controlling the proteasomal degradation of PIM kinases (Figure 2A).

Next, we assessed whether the ubiquitination of PIM1/2 was also sensitive to DUB inhibition using the previously described ubiquitination assay. To this end, cells were pretreated with DMSO or PR-619 for 30 min prior to the addition of MG-132, and lysates were collected at 2 and 4 h. Immunoblotting for ubiquitin after immunoprecipitation revealed that PIM1 and PIM2 were more highly ubiquitinated in cells treated with PR-619, providing further evidence that a DUB is responsible for regulating the ubiquitination and degradation of PIM kinases (Figure 2B). Based on the literature, we identified four DUBs that have been associated with hypoxia: USP13, USP28, USP46, and CYLD [32]. To determine whether any of these candidates affected PIM1/2 protein levels, we transfected each into PC3-LN4 cells and monitored PIM1/2 expression by western blotting. While the ectopic overexpression of several DUBs increased PIM1/2 levels, USP28 caused the greatest increase (Figure 2C). Importantly, USP28 is not inhibited by HBX 41108 [44], which explains why PR-619 decreased PIM levels but HBX 41108 did not. Therefore, we explored the potential of PIM1 and PIM2 as substrates of USP28.

### 3.3. USP28 Increases PIM Stability

Because DUBs stabilized their target proteins, we assessed the effect of USP28 overexpression on PIM kinase stability. To this end, 293T cells transfected with a control vector or USP28 were treated with CHX, lysates were collected at the indicated time points, and the half-lives of PIM1 and PIM2 were assessed by western blotting and densitometry. In normoxia, the half-life of PIM2 in cells transfected with the control vector was 1.26 h, whereas the half-life of PIM2 in cells transfected with USP28 was 7.3 h, indicating that the overexpression of USP28 significantly increased PIM2 stability (*p* = 0.01). We observed similar results with PIM1 (0.98 h vs. 1.95 h) (Figure 3A). We repeated this experiment in 1% O_2_ and observed that USP28 overexpression led to even greater stabilization of PIM1 and PIM2 (vector vs. USP28: PIM1, 2.00 h vs. 5.37 h, *p* = 0.01; PIM2, 1.73 h vs. 7.45 h, *p* = 0.007) (Figure 3B).

Next, we created USP28-knockdown cells (shUSP28) by transducing PC3-LN4 cells with two different shRNAs against USP28. Control cells were transduced with the empty vector. The knockdown of USP28 was sufficient to block the induction of PIM1/2 in response to hypoxia, suggesting that USP28 is required to regulate PIM1/2 expression in hypoxia (Figure 3C). It is of note that total levels of USP28 were not altered by hypoxia, suggesting an increase in activity toward PIM instead of general USP28 upregulation (Figure 3C). Because shRNA #1 displayed a stronger knockdown of USP28, we used this shRNA for further experiments. We next assessed the effect of USP28 knockdown on PIM1/2 ubiquitination. Control or shUSP28 PC3-LN4 cells were transfected with HA-PIM1 or HA-PIM2, treated with MG-132, and harvested at 2 or 4 h, after which PIM isoforms were immunoprecipitated. PIM1 and PIM2 ubiquitination was significantly increased in cells lacking USP28 (Figure 3D). Taken together, these results indicated that USP28 is sufficient to regulate the stability of PIM kinases, regardless of oxygen tension, and necessary for the induction of PIM kinases in response to hypoxia.

### 3.4. USP28 Interacts with PIM Kinases Preferentially in Hypoxia

Because we observed no increase in USP28 levels in hypoxia, we hypothesized that hypoxia might increase the affinity of USP28 for PIM kinases. To examine this, we performed co-immunoprecipitation to determine whether these proteins preferentially interact in hypoxia. 293T cells were transfected with HA-PIM1/2 and USP28-Myc and incubated in normoxia or hypoxia for 1 or 6 h prior to harvest. HA-PIM1/2 were immunoprecipitated, and USP28 interaction was monitored by blotting for Myc. Interestingly, USP28 was only bound to PIM1 and PIM2 in hypoxia, and this binding occurred as early as 1 h (Figure 4A,B). Hence, the induction of PIM kinases in hypoxia can be attributed to increased interaction with USP28 and subsequent deubiquitination. We also examined the effect of protein kinase B (Akt) inhibition on this interaction, as the E3 ubiquitin ligase most commonly associated with USP28—FBW7—is regulated by glycogen synthase kinase 3β (GSK-3β) through Akt [45]. The inhibition of Akt activity did not affect the binding of USP28 and PIM2 in normoxia or hypoxia (Figure 4C).

### 3.5. USP28 Regulates PIM Protein Levels In Vivo

Finally, we performed in vivo tumorigenesis assays to confirm the relevance of this signaling axis in tumors and further investigate the role of USP28 in tumor growth. Five million control or shUSP28 PC3-LN4 cells were injected subcutaneously into the flanks of immunocompromised mice. We previously observed that treatment with sunitinib (an inhibitor of vascular endothelial growth factor [VEGF] signaling) results in hypoxia and significantly increases PIM1 levels [14]. Therefore, we treated both cohorts with a vehicle or sunitinib once tumors were established. In the vehicle-treated mice, control tumors grew more rapidly than shUSP28 tumors, indicating that USP28 promotes tumor growth in this prostate cancer model, potentially by inducing PIM1/2 expression. Although we did not observe a significant difference in the tumor volume, shUSP28 tumors tended to be smaller than control tumors, and sunitinib was able to further decrease the size of these tumors (Figure 5A). This effect mimics previous findings from our group showing that a combined inhibition of PIM and VEGF signaling produces an enhanced antitumor activity [14]. At the end of the study, tumors were harvested to assess PIM1 levels in each cohort. The western blotting analysis of four individual tumors showed a significant reduction in PIM1 in tumors lacking USP28 (Figure 5B). The immunohistochemical staining of PIM1 confirmed this result, showing a significant decrease in PIM1 in shUSP28 tumors compared to that in control tumors (Figure 5C). Moreover, we observed a dramatic increase in PIM1 in sunitinib-treated tumors compared to that in vehicle-treated tumors, whereas there was only a modest increase in PIM1 levels following sunitinib treatment in the shUSP28 tumors that was equivalent to the levels in untreated controls, suggesting that the hypoxic induction of PIM1 is highly sensitive to the loss of USP28 (Figure 5C).

## 4. Discussion

PIM kinases play important protumorigenic roles in multiple cancer types [5]. They are particularly important in prostate cancer, where they are commonly upregulated [4]. This upregulation is of particular interest, because the prostate gland is highly hypoxic [46]. Although our group and others previously observed that PIM1 levels are increased in hypoxia, the mechanism underlying this phenomenon has never been described. This increase in PIM kinases in hypoxia allows cancer cells to survive hypoxic stress [11], including reactive oxygen species, which are increased in hypoxia [47]. Being able to respond to hypoxia is vital for tumor cells, since tumors rapidly outgrow their blood supply as they proliferate. This leads to decreased oxygen throughout the tumor, and the tumor must respond by promoting angiogenesis or new blood vessel growth, which provides both oxygen and nutrients to the tumor. Our previous work showed that PIM kinases can promote tumor angiogenesis [14]. Here, we identified an association between PIM kinases and the DUB USP28. USP28 increased PIM1 and PIM2 protein stability and interacted with PIM1/2 preferentially in hypoxia (Figure 5D). Our results showed that USP28 was necessary for the increase in PIM observed in hypoxia both in vitro and in vivo.

Unlike most protein kinases, PIM kinases do not contain any regulatory domains and are constitutively active upon translation. Therefore, characterizing the mechanisms that control PIM levels is critically important for understanding how these kinases are dysregulated in cancer. Previous studies have largely focused on the transcriptional regulation of PIM, namely via JAK/STAT signaling. In contrast, we found that hypoxia did not change the levels of *PIM1*, *PIM2*, or *PIM3*, suggesting that hypoxia impacts PIM at the post-translational level. This is somewhat uncommon in hypoxia, as a vast majority of hypoxia-induced proteins can be attributed to HIF-1 transcriptional upregulation, including factors that promote angiogenesis, such as VEGF and angiopoietin-like 4 [48,49], or relieve the deleterious effects of hypoxia, such as HK2 and heme oxygenase 1 [50,51]. Although we observed HIF-1 target genes upregulated at the transcriptional level in hypoxia, we did not observe any significant increase in PIM kinase transcript levels (Figure 1).

Instead, we showed that hypoxia altered the ubiquitination and proteasomal degradation of PIM1/2 and that this effect was dependent upon the activity of DUBs. A screen of DUBs that are associated with hypoxia led us to identify USP28 as a key factor in controlling PIM protein stability. The overexpression of USP28 increased PIM1/2 stability, whereas the knockdown of USP28 decreased PIM1/2 levels and increased their ubiquitination. The tight regulation of the deubiquitination process is an important mechanism by which hypoxic cells can regulate their protein complement without the high cost of new translation [32]. Because of their low oxygen tension, hypoxic cells are unable to undergo oxidative phosphorylation. This is particularly true of hypoxic cancer cells. Therefore, being able to rescue specific factors from proteasomal degradation can save cells from having to expend the energy to translate proteins anew. This process has been best studied in the regulation of HIF-α subunits themselves. In addition to the loss of ubiquitination due to the inactivation of prolyl hydroxlyases, some DUBs, including USP28, have been shown to deubiquitinate HIF-α subunits, increasing their protein concentration and stimulating the subsequent transcriptional response [32]. However, most DUBs that have been shown to be increased in hypoxia are increased at the transcriptional level downstream of HIF-1 activation [32]. Conversely, USP28 activity has been shown to be differentially regulated in hypoxia by SUMOylation [52], suggesting that USP28 might be particularly active in hypoxia.

Mechanistically, USP28 preferentially bound to PIM1/2 in hypoxia, suggesting that it is recruited to PIM kinases particularly under low oxygen tension. USP28 is usually recruited to its substrates through interaction with an E3 ubiquitin ligase [38], most commonly FBW7. However, we did not observe any interaction between FBW7 and PIM kinases (Appendix A). Further, many FBW7 targets are phosphorylated by GSK-3β, which has been shown to be a direct target of Akt [53]. However, Akt inhibition, which led to active GSK-3β (i.e., no S9 phosphorylation), did not affect the interaction of USP28 with PIM2 (Figure 4C), suggesting that a different E3 ubiquitin ligase is responsible for ubiquitinating PIM kinases and recruiting USP28. Previous studies have described the recruitment of USP28 by the E3 ubiquitin ligases kelch-like family member 2 (KLHL2) [54] and ring finger and CHY zinc finger domain containing 1 (RCHY1) [55], but little is known about how these factors are affected by tumor hypoxia. KLHL2 has mainly been studied in hypertension [56], another disease in which PIM kinases play key roles and their overexpression is associated with poor prognosis [57]. This is intriguing, as we have previously shown PIM kinases to be necessary for the induction of new blood vessel formation in prostate cancer [14]. There is no literature on RCHY1 in hypoxia, but previous studies have shown that it may be involved in prostate carcinogenesis. For instance, RCHY1 interacts with the androgen receptor to promote target gene expression [58] and promotes the degradation of p53 [59]. These ligases and others may regulate hypoxia-inducible proteins in prostate cancer through USP28. Identifying the E3 ligase associated with the USP28-PIM axis will help clarify the underlying biology of prostate cancer.

In conclusion, we identified the DUB USP28 as a novel regulator of PIM stability in hypoxia. This hypoxia-induced pathway plays vital roles in tumor progression, so identifying factors regulating this pathway is important for understanding the underlying tumor biology.

## Figures and Tables

**Figure 1 cells-11-01006-f001:**
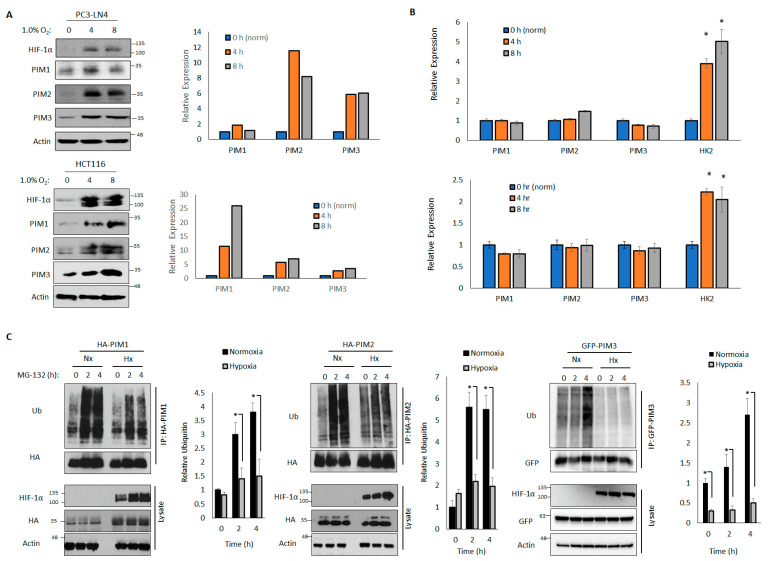
Hypoxia stabilizes Moloney murine leukemia virus (PIM) kinases by reducing ubiquitination and proteasomal degradation. PC3-LN4 and HCT116 cells were cultured in normoxia (21% O_2_) or hypoxia (1% O_2_) for 4 or 8 h. (**A**) Lysates were harvested for western blotting (right: quantification). (**B**) RNA was harvested for quantitative reverse-transcriptase polymerase chain reaction. Beta-actin was used as a control. (**C**) 293T cells were transfected with HA-PIM1, HA-PIM2, or GFP-PIM3 and cultured in normoxia or hypoxia for 30 min. Cells were treated with MG-132 for 2 or 4 h to block protein degradation. HA-PIM1/2 and GFP-PIM3 were immunoprecipitated, and immunoprecipitated and input lysates were used for western blotting. Right, quantification of relative ubiquitin density. * *p* < 0.05.

**Figure 2 cells-11-01006-f002:**
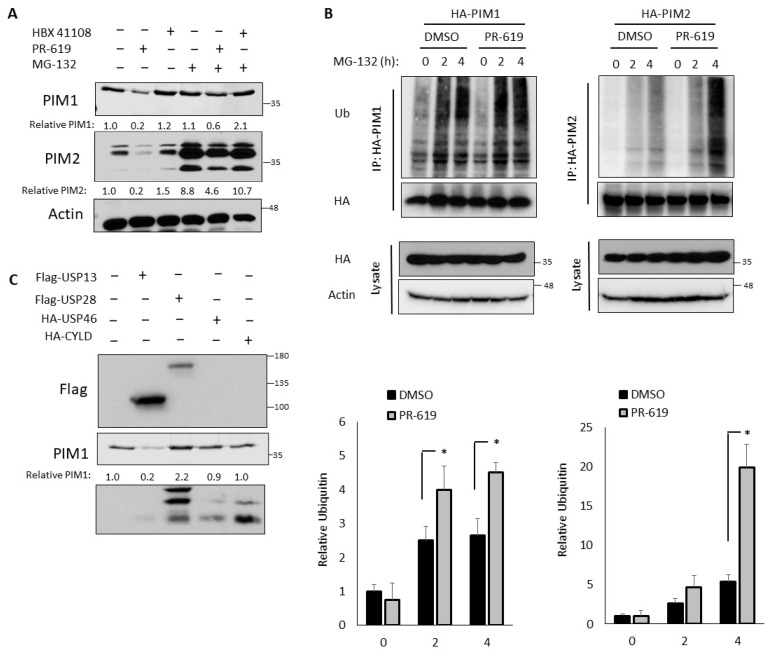
PIM kinase ubiquitination and stability is controlled by deubiquitination by ubiquitin-specific protease 28 (USP28). (**A**) PC3-LN4 cells transfected with HA-PIM1 were treated with PR619, HBX 41108, MG-132, or a combination for 6 h. Lysates were harvested for western blotting. (**B**) 293T cells were transfected with HA-PIM1 or HA-PIM2 and pretreated with vehicle or PR619 for 30 min. Then, cells were treated with MG-132 for 2 or 4 h to block protein degradation. HA-PIM1/2 were immunoprecipitated, and immunoprecipitated and input lysates were used for western blotting. Below, quantification of relative ubiquitin density. (**C**) PC3-LN4 cells were co-transfected with HA-PIM1 and a control vector, Flag-USP13, Flag-USP28, HA-USP28, or HA-CYLD. Lysates were harvested for western blotting approximately 16 h later. * *p* < 0.05.

**Figure 3 cells-11-01006-f003:**
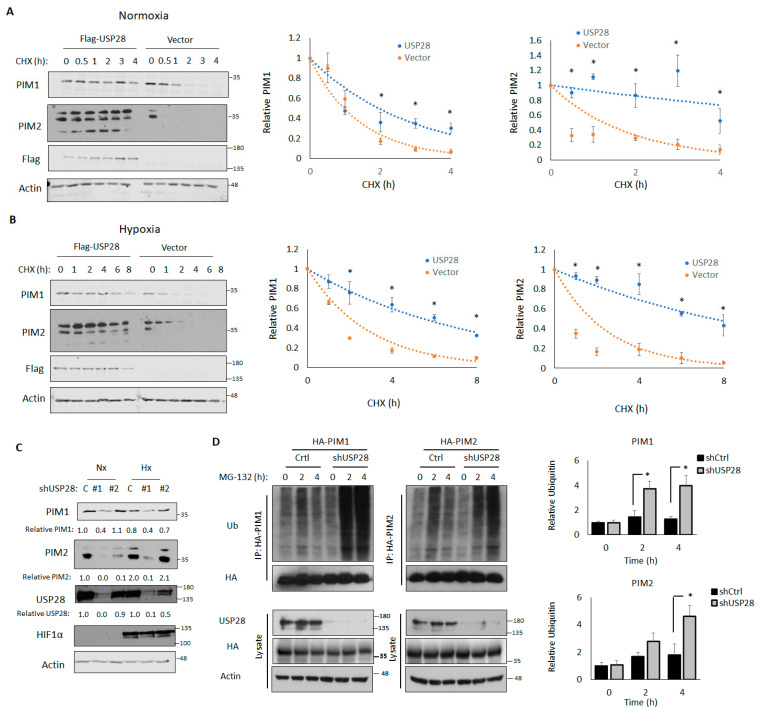
USP28 controls PIM1/2 stability in normoxia and hypoxia. 293T cells were co-transfected with HA-PIM1 and Flag-USP28 or a control vector. Cells were then treated with 10 µM cycloheximide (CHX) for 30 min, 1, 2, 3, or 4 h in normoxia (**A**) or 1, 2, 4, 6, or 8 h in hypoxia (**B**), and lysates were harvested for western blotting. Middle, quantification of PIM1; right, quantification of PIM2. Quantifications represent the results of three independent experiments. Error bars indicate the standard error of the mean. (**C**) PC3-LN4 USP28-knockdown cells were created by transducing PC3-LN4 cells with USP28 short hairpin RNAs. Control cells were transduced with a control vector. Control and USP28-knockdown cells were transfected with HA-PIM1 and cultured for 6 h in hypoxia. Lysates were harvested for western blotting. (**D**) PC3-LN4 control and USP28-knockdown cells were transfected with HA-PIM1 or HA-PIM2 and treated with MG-132 for 2 or 4 h to block protein degradation. HA-PIM1/2 were immunoprecipitated, and immunoprecipitated and input lysates were used for western blotting. Right, quantification of relative ubiquitin density. * *p* < 0.05.

**Figure 4 cells-11-01006-f004:**
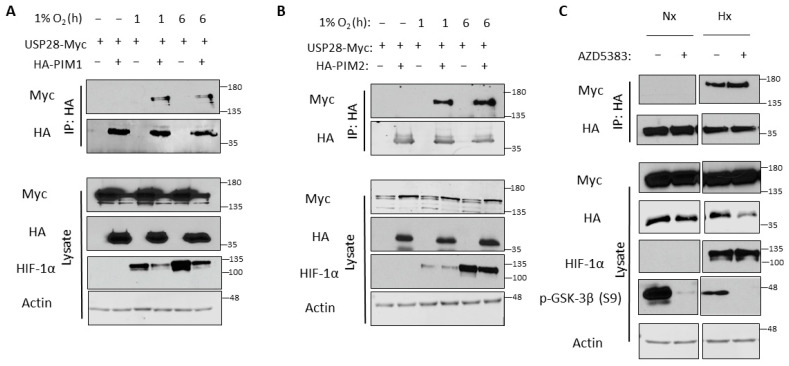
USP28 binds to PIM1/2 preferentially in hypoxia. 293T cells were co-transfected with USP28-Myc and HA-PIM1 (**A**) or HA-PIM2 (**B**) and cultured in normoxia or hypoxia for 1 or 6 h. HA-PIM1/2 were immunoprecipitated, and immunoprecipitated and input lysates were used for western blotting. Cells transfected with USP28-Myc alone were used as a negative control. (**C**) 293T cells were co-transfected with HA-PIM2 and USP28-Myc, treated with vehicle or an Akt inhibitor (AZD5383), and cultured in normoxia or hypoxia for 6 h. HA-PIM2 was immunoprecipitated, and immunoprecipitated and input lysates were used for western blotting.

**Figure 5 cells-11-01006-f005:**
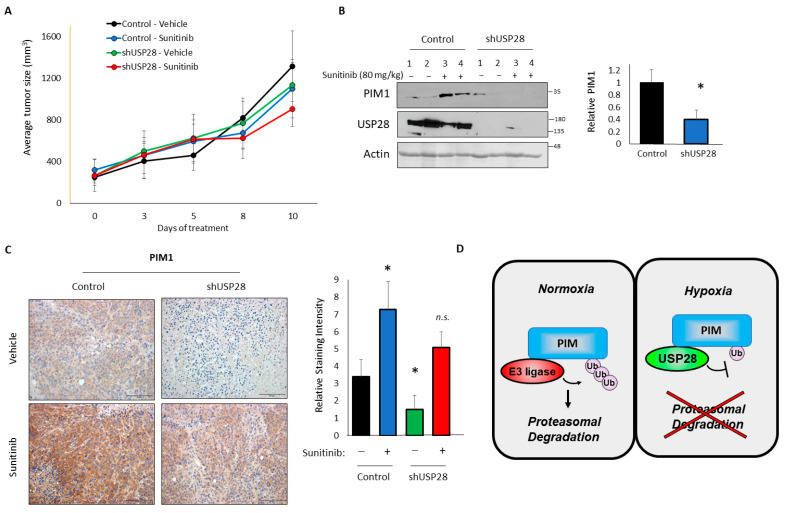
Upregulation of PIM1 in response to anti-angiogenic agents requires USP28. Mice were injected subcutaneously with PC3-LN4 control or USP28-knockdown cells and treated with a vehicle or sunitinib. (**A**) Growth curve of tumors in the four groups (n = 8). (**B**) Tumors were harvested, and lysates were used for western blotting. Right, quantification. (**C**) Representative immunohistochemical staining of PIM1 in tumors from the four groups (scale bar = 50 μM). Right, quantification. (**D**) Model. * *p* < 0.05.

**Table 1 cells-11-01006-t001:** Primer sequences used for quantitative polymerase chain reaction.

Target	Primer Sequence
PIM1	CGACATCAAGGACGAAAACATC ACTCTGGAGGGCTATACACTC
PIM2	GAACATCCTGATAGACCTACGC CATGGTACTGGTGTCGAGAG
PIM3	GACATCCCCTTCGAGCAG ATGGGCCGCAATCTGATC
HK2	Purchased from Qiagen (catalog # QT00013209)
ACTB	TGACGTGGACATCCGCAAAG CTGGAAGGTGGACAGCGAGG

ACTB, beta-actin; HK2, hexokinase 2; PIM, proviral integration site for Moloney murine leukemia virus.

## Data Availability

Not applicable.

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
