# Peer review of "Stabilization of PIM Kinases in Hypoxia Is Mediated by the Deubiquitinase USP28"

_cells, 2022, doi:10.3390/cells11061006_

Round 1

Reviewer 1 Report

The authors are tackling an interesting and relevant question on how the stability of the oncogenic PIM proteins is regulated. They have now observed that the USP28 deubiquitinase, which is known to support stability of proteins such as HIF1a and MYC under hypoxic conditions, also stabilises PIM kinases, at least PIM1 and PIM2. The story would have been more complete, if also interaction with PIM3 had been shown, and if the E3 ligase involved had been identified, as now there is only some speculation on potential candidates.

One of the major weaknesses is the usage of different cell lines in every other experiment, confusing the reader. Why to use 293T cells, as also PC3 cells are well transfectable and produce more relevant results? As such, it is good to confirm results in more than one cell line, but the HCT116 cells would have been a better option there. While interesting results are presented, there are several experiments, the results of which are not yet convincing or conclusive enough, as described in detail below. Therefore, a proper revision of the manuscript is recommended before its publication.

Introduction:

  • Ref. 3 is not the one where the the basic properties of PIM kinases have been reported, so please refer either to original publications (e.g. Ref. 20) and/or a review, where it is also described how cytokines induce expression of PIMs, which in turn regulate cytokine signaling by several ways. Also Ref. 8 is just a follow-up paper for other previous publications describing the anti-apoptotic effects of PIM kinases, so reference to a review would be better also here. 
  • What do the authors mean with their strange statement of MYC transcription factors being closely related to PIM kinases (lines 91-92, ref. 34)? These proteins collaborate in cancer cells, but there is no homology between them.
  • There are multiple acronyms in the text, which should be explained to facilitate reading.
  • As the authors are not the only or even first ones, who have reported PIM kinases to support tumor angiogenesis, some other refs should be added. In the discussion, the authors admit that also others have previously observed that PIM1 levels are increased in hypoxia, so it would be nice to add refs also to some of those there or in the Introduction.

Materials and methods:

  • One should describe the nature of the different cell lines, and also mention that PC3-LN4 cells represent a more metastatic clone of the original PC3 prostate cancer cell line. How reliable conclusions can one make on the applicability of the results from 293T cells as compared to PC3 cells? To avoid any confusion, one should clearly label each blot with the name of the cell line used in each case.

Results:

  • All blots need molecular weight markers, and many also need proper quantitation.
  • Fig. 1A: The data should be quantitated. Based on the blots, levels of all three PIM proteins are upregulated by hypoxia, so leaving out PIM3 from all subsequent experiments is not well justified, especially as in solid tumors, like prostate cancer, PIM1 and PIM3 play more significant roles than PIM2.
  • Fig. 1C: Why was this experiment not carried out in the same cells as in 1A and B, which would have been more relevant? It would have been good to include also PIM3 here. After MG132 treatment, there is clearly increased accumulation of ubiquitin under normoxia as compared to hypoxia, but would one not expect to see a difference already in the absence of MG132? Also the authors state that “the rates and total amounts of ubiquitination of PIM1 and PIM2 were significantly reduced in hypoxia compared to normoxia at all time points examined”, but no significant differences were observed at the 0h time-point. Furthermore, the details of this experiment and its timing should be described in the Materials and methods.
  • Fig. 2A: Can one rule out the other possibility that in addition to increased DUB activity, there is also decreased E3 ligase activity? Quantitation of the blots needed.
  • Fig. 2B: Why now again switched to 293T cells? No difference -/+ PR-619 to start with, unlike in Fig. 2A with PC3 cells?
  • Fig. 2C: Does HBX 41108 inhibit the other three DUBs (not USB28)? Ref? Quantitation of this blot needed to support conclusions, as the effects on PIM1 were minor. However, PIM1 was ectopically expressed, while PIM2 only endogenous, making it more difficult to draw conclusions. One could save space by cutting the HA blot to two halves.
  • Fig. 3A&B: USP28 overexpression clearly increases PIM stability, but the blots and their quantitations do not seem to correlate well with each other or with data shown in Fig. 1. This may be partly due to the use of 293T cells instead of the more relevant PC3 cells. It is also possible that the stability of an ectopically overexpressed PIM protein is not regulated in the same fashion as the endogenous PIM proteins.
  • Fig. 3C: How come PIM1 protein is not detected in the first lane, but is seen with shUSP28 #2 cell samples? And why PIM2 levels do not correlate with USP28 levels?
  • Fig. 4: Why are A and B not shown in an alphabetical order? If USP28 binds to PIMs only in hypoxia, how does its overexpression stabilize PIMs also in normoxia, as shown in Fig. 3? It should be noted that also PIM kinases can phosphorylate GSK3B, even though GSK3B does not seem to play a role here. However, AKT inhibition seems to reduce PIM2 levels in hypoxia, even though the interaction between PIM2 and USP28 is not affected.
  • Fig. 5A: The authors state that the “shUSP28 321 tumors responded better to sunitinib than control tumors did”, but based on the graph there was not any major difference in the reduction of tumor size by sunitinib?
  • Fig. 5B: There was quite a lot of variation in the PIM1 and PIM2 levels between the only four parallel samples analysed, so it is hard to draw any conclusions from there. How about PIM levels in sunitinib-treated samples? Or were those included in the blot, as according to Materials and methods, only 8 mice were used altogether?
  • Fig. 5D: These data look more convincing than those for Fig. 5B, so maybe the blots could be left out.
  • Suppl. Fig. 1: One should include a positive control to confirm that the reagents otherwise worked.

Reviewer 2 Report

In the paper entitled: "Stabilization of PIM1/2 kinases in hypoxia is mediated by the deubiquitinase USP28" the authors showed that the deubiquitinase USP28 blocks the ubiquitination and increases the stability of PIM1/2. The topic of the paper is interesting for the researchers in the field. the authors have to better explain some points. 

  • the authors shoyld to explain why they choose 0-2-4h of hypoxia after the first western blot experiment in which they use 0-4-8h of hypoxia condition
  • the have tpo add marker molecular weight to all the western blot membranes
  • may they clarify the use of USP7 inhibitor?
  • in the experiments carried out to demontrate that PIM kinases are regulated by DUBs (Fig.2) the authors show the results in normoxia conditions but what happen in hypoxia?moreover, in figure 2A and B could they quantify the western blot analysis?
  • The authors in this paper can better explain why do they use different cell lines in the experiments?
  • In the figure 3c they may quantify the total level of USP28 to support their sentences about the inaltered USP28 protein levels between normoxia and hypoxia condition
  • Figure 4 could the authors put it in correct order (A-B-C)

Reviewer 3 Report

In this  manuscript, the authors determined USP28 as a deubiquitinase regulating the stability of PIM1/2 in hypoxia. Cellular ubiquitination and degradation assay were performed to demonstrate the role of USP28 in vitro. Also, using the xenograft model, the authors showed that USP28 enhances tumor growth and PIM1 level in vivo. Overall the manuscript is well organized and easy to understand.

Major concern:

  1. Line 231: Can the authors explain why MG-132 blocks the effect of PR-619?

Although MG-132 inhibits the activity of proteasome, the increased ubiquitinated PIM species is very likely to remain unchanged, which results in the decrease in level of the unmodified species in the presence of PR-619.

  1. What are the multiple bands in the bolts of PIM2? Are these bands different isoforms ? If not, can the authors indicate the true band of PIM2?
  2. The WB signal of PIM2 seems inconsistency between assays using the same cell line. In the Figure 2A and 3C, the mock treated PC3-LN4 cells showed variations in the expression of PIM2. The same concern exists in the PIM2 signal of Figure 2C and 3A.
  3. Can the authors discuss why USP28 overexpression inhibits the degradation of PIM1/2 in normoxia even if no interaction was detected between USP28 and PIM1/2. It will be really interesting to elucidate how oxygen regulates this interaction.
  4. Why hypoxia failed to induce the level of PIM1/2 in Figure 4A and 4B?

Minor concern:

  1. The figure legend should provide necessary experimental details such as how long the cells were treated with inhibitors.
  2. The CHX chase assay may need more replicates to get an accurate half-life.
  3. The HIF1a blot in Figure 4A need to be adjusted for alignment.

Round 2

Reviewer 1 Report

Even though it would have been nice to add some more data from suggested experiments, most of my comments have now been addressed to a sufficient extent, so the manuscript can be recommended for publication after two minor corrections:

The sentence on cooperation rather than relationship between PIM and MYC in tumorigenesis has now been correctly formulated. However, Ref. 39 is not correct, as it does not describe this, so please replace it with earlier refs from Berns group and/or later ones on prostate cancer from Abdulkadir group.

The description on data shown in Fig. 5A should be clarified in the text, as sunitinib seems to reduce tumor growth to a very similar extent -/+ shUSP28.

Author Response

We thank the reviewer for their comments and have made the suggested changes accordingly

Comments 1: The sentence on cooperation rather than relationship between PIM and MYC in tumorigenesis has now been correctly formulated. However, Ref. 39 is not correct, as it does not describe this, so please replace it with earlier refs from Berns group and/or later ones on prostate cancer from Abdulkadir group. 

Response: Per your suggestion, these have been updated to reference the cooperation between PIM and Myc in prostate cancer.

Comments 2: The description on data shown in Fig. 5A should be clarified in the text, as sunitinib seems to reduce tumor growth to a very similar extent -/+ shUSP28.

Response: Thank you for clarifying this concern. The text has been revised to indicate USP28-knockdown tumors treated with sunitinib had the smallest tumor volume of the four groups (lines 244-246). 
